# Coupling Attention and Memory: A Dynamic Memory Module for Efficient Adaptation with Pretrained LLMs

## Abstract

Pretrained large language models (LLMs) are highly capable but still require adaptation for various domains. Existing fine-tuning strategies typically assume either access to all target task data *simultaneously* (*e.g.,* multi-task learning), or a sequential data stream, as in continual learning, where the former tackles the simultaneous task interference issue while the latter focuses on addressing the catastrophic forgetting problem. In this work, we propose a unified approach to address both scenarios. We present DynMem, a unified framework that tackles both settings with a lightweight dynamic memory module built on top of frozen pretrained LLMs. DynMem encodes past examples into a fixed-sized memory bank. We design a novel dynamic update mechanism where new examples and existing memory entries are ranked based on their *accumulated* attention scores, and the lowest-ranked examples are thus pruned to maintain size. To further reduce recency bias, we adopt a new bi-level memory design: $L_1$ Memory is actively used by the backbone LLM, while $L_2$ Memory stores more diverse examples for improved effectiveness at minimal cost. The design also supports more flexible test-time scaling by allowing larger memory banks. We evaluate DynMem under both simultaneous and continual learning settings. Our method consistently outperforms state-of-the-art baselines tailored for each scenario, demonstrating its great potential in mitigating task inference for both simultaneous and sequential learning. In particular, DynMem outperforms the state-of-the-art method in simultaneous adaptation across different models, yet achieves this with approximately 50% fewer trainable parameters.

## 1 Introduction

The paradigm of pre-trained large language models (LLMs) has established a powerful foundation for artificial intelligence (Achiam et al., 2023; Bai et al., 2023; Dubey et al., 2024), yet their ability of dynamic adaptation remains a critical frontier. Thus, many researchers have attempted to design more efficient fine-tuning strategies. Recent parameter-efficient fine-tuning (PEFT) methods, *e.g.,* LoRA (Hu et al., 2022) and prompt tuning Lester et al. (2021), typically focus on updating a small amount of extra model parameters to learn a single or multiple tasks jointly. However, they are limited by assuming access to all data simultaneously, *i.e., simultaneous adaptation*.

To address the more realistic sequential data stream scenario, various parameter-efficient continual learning methods (Zhu et al., 2022; Chen et al., 2023; Zhao et al., 2024) are developed for resolving the catastrophic forgetting issue. These methods prevent catastrophic forgetting by allocating separate, architecturally disjoint parameters (*e.g.,* soft prompts or adapters (Poth et al., 2023)) for each task. However, this isolationist approach introduces critical limitations. It creates a task-agnostic inference problem, as it requires an oracle to select the correct parameters at test time, and inherently restricts forward transfer by siloing knowledge within each module (Zheng et al., 2024). More critically, their effectiveness in simultaneous adaptation (*e.g.,* multi-task learning) is largely unknown. In this paper, we aim to bridge this gap and design a unified method for both scenarios.

Inspired by recent memory-augmented methods (Yang et al., 2024; Wu et al., 2022a; Zhai et al., 2025; Mitchell et al., 2022), we propose DynMem, a lightweight memory module operating on a

fixed-size memory bank that enables efficient LLM adaptation across both sequential and simultaneous learning settings. Following recent work, DYNMEM encodes past training examples into vectors that are stored in a dynamic memory bank. To support more efficient reading and forgetting mechanisms, we introduce a bi-level memory system that utilizes the attention scores from the frozen backbone LLM. First, a compact $L_1$ *Memory* maintains a compact set of recently retrieved, highly relevant samples, which are integrated into LLM via a gated fusion module. However, this can introduce recency bias, potentially degrading long-term knowledge retention. To mitigate this, we incorporate a larger $L_2$ *Memory*, which caches more historically high-ranking samples. Since $L_2$ Memory does not directly interact with the LLM, it significantly enhances the method's effectiveness at minimal cost. All samples in both memory stores are periodically ranked based on an attention-based ranking mechanism, where those lowest-ranked samples are dequeued to leave space for new samples. In other words, both memory reading and pruning operations are based on the attention module, which is jointly trained in an end-to-end fashion. During inference, DYNMEM employs a cross-stage retrieval process, integrating the selected memories with the current input via a gated fusion mechanism, allowing the model to dynamically leverage past knowledge for the task at hand.

We conduct a comprehensive empirical evaluation of DYNMEM across two primary adaptation scenarios: Continual Adaptation, which assesses the model's resilience to forgetting and its capacity for forward transfer, and Simultaneous Adaptation, which includes two distinct sub-settings, Single-task Tuning and Multi-task Integration. Across this diverse suite of benchmarks, DYNMEM demonstrates significant gains in knowledge retention, adaptation, and generalization, establishing a new state-of-the-art in versatile model adaptation. To summarize, our contributions include:

- We propose DYNMEM, a lightweight dynamic memory module for efficient adaptation with pretrained LLMs. As far as we know, we are the first to unify simultaneous and continual learning paradigms within a single, cohesive architecture.

- We introduce a unique bi-level memory system, featuring a working and reserved memory managed by an attention-based filtering mechanism, which enables efficient, example-level knowledge retention and retrieval with a compact memory size.

- We conduct a comprehensive empirical evaluation across a diverse suite of benchmarks spanning continual, single-task, and multi-task adaptation. Our results demonstrate that DYNMEM achieves state-of-the-art results across the board, significantly outperforming specialized methods in their respective domains.

## 2 PRELIMINARIES

Our work aims to develop a single, unified framework that excels in two distinct paradigms of model adaptation. To establish the context for this approach, we first formalize these paradigms below.

**Continual Adaptation.** Commonly known as Continual Learning, continual adaptation addresses the more dynamic scenario where tasks arrive sequentially, $\mathcal{T}_1, \mathcal{T}_2, \ldots, \mathcal{T}_N$. When training on the current task $\mathcal{T}_k$, the model has only access to its corresponding dataset $\mathcal{D}_k$, and data from past tasks $\{\mathcal{D}_1, \ldots, \mathcal{D}_{k-1}\}$ is unavailable. The main challenge in this setting is *catastrophic forgetting*, defined as the severe degradation of performance on previously learned tasks after the model updates for new ones. Formally, let $A(\theta_j; \mathcal{D}_i)$ denote the accuracy of the model with parameters $\theta_j$ (having learned up to task $j$) on the dataset for task $i$. Forgetting is measured by the performance drop from $A(\theta_i; \mathcal{D}_i)$ to $A(\theta_k; \mathcal{D}_i)$ for any $i < k$. The objective here is twofold: learn the new task effectively while simultaneously preserving knowledge from all previously seen tasks.

**Simulteneous Adaptation.** In this paradigm, a model $f_\theta$ is assumed to have full access to the complete datasets for a set of $N$ tasks, $\{\mathcal{D}_1, \ldots, \mathcal{D}_N\}$. The primary goal is to learn a single set of parameters $\theta$ that performs well across these tasks by leveraging their shared structure. This paradigm encompasses several key evaluation settings: a) *Single-task Tuning* The model is specialized for a single task $\mathcal{T}_i$ by fine-tuning exclusively on its dataset $\mathcal{D}_i$. b) *Multi-task Integration*: A single model is jointly trained on the union of all task datasets, $\mathcal{D}_{\text{all}} = \bigcup_{i=1}^{N} \mathcal{D}_i$, to encourage knowledge sharing.

# 3 THE DYNMEM FRAMEWORK

Here, we introduce DYNMEM, a framework designed to unify simultaneous and continual adaptation. At a high level, DYNMEM augments a pre-trained LLM, denoted as a parametric function $f_\theta$, with a dynamic, bi-level memory system as shown in Figure 1. Unless otherwise specified, the backbone LLM remains frozen throughout the paper.

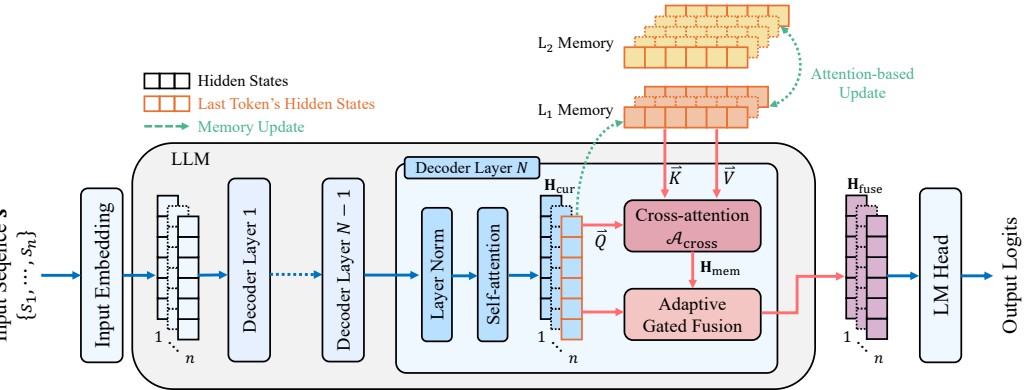

Figure 1: The overall architecture of DYNMEM. During the training phase, as new tasks arrive, the model is continuously fine-tuned and example representations are generated based on the final layer hidden states of the backbone LLM. Note that the model only interacts with the $L_1$ Memory via cross-attention and a gated fusion layer to produce predictions (§3.1). The attention scores from this interaction also guide the periodic pruning of $L_1$ and $L_2$ to maintain a compact memory size.

Similar to recent memory-augmented methods, we derive memory entries using the backbone LLM as the encoder. The novelty of DYNMEM lies in its bi-level memory system, designed to balance immediate task relevance with long-term knowledge diversity. This memory module is strategically inserted at the final decoder layer of the LLM, intercepting the layer's output hidden states to perform memory interaction before the final prediction. The system consists of a $L_1$ Memory ($\mathcal{M}_{L_1}$) for high-relevance, active samples and a larger ($L_2$) Memory ($\mathcal{M}_{L_2}$) that serves as a long-term reservoir. Both memory caches hold a predefined capacity, with $|\mathcal{M}_{L_2}| > |\mathcal{M}_{L_1}|$.

The basic entry in both memories is a vector $\mathbf{m}_e$ representing a single training example $(x_e, y_e) \in \mathcal{D}_i$, where $x_e$ is the task prompt input and $y_e$ is the corresponding gold output. Consistent with the module's placement, this vector is generated by extracting the semantically rich hidden states of the final token from the last decoder layer, *i.e.,* $\mathbf{m}_e = \text{Decoder}_{\text{last}}(f_\theta(\text{Concat}(x_e, y_e)))_{[-1]} \in \mathbb{R}^d$, where $d$ is the hidden dimensionality. This representation effectively summarizes the input-output mapping for a given example and forms the candidate pool for our memory bank.

## 3.1 MEMORY INTERACTION AND INTEGRATION

The flexible reading and forgetting interactions on memory are achieved by three key modules: a cross-attention module for retrieving knowledge, an update strategy for maintaining the memory bank, and a gated layer for integrating retrieved information.

**Attention-based Knowledge Retrieval.** To retrieve and integrate relevant memory knowledge, we design a cross-attention module $\mathcal{A}_{\text{cross}}$. Denote the hidden state representation of the current input from the final decoder layer as $\mathbf{H}_{\text{cur}} \in \mathbb{R}^{L \times d}$. During the model's forward pass, this representation interacts exclusively with the contents of the $L_1$ Memory, whose vectors $\{\mathbf{m}_1, \cdots, \mathbf{m}_{|\mathcal{M}_{L_1}|}\}$ are concatenated into a matrix $\mathbf{M}_{L_1}$. To leverage the LLM's pre-trained weights and ensure parameter efficiency, we initialize this cross-attention module from the self-attention block of the same decoder layer. Specifically, the input hidden states $\mathbf{H}_{\text{cur}}$ form the query, while the memory vectors in $\mathbf{M}_{L_1}$ form the key and value pairs: $\mathbf{Q} = \mathbf{H}_{\text{cur}} W^Q$, $\mathbf{K} = \mathbf{M}_{L_1} W^K$, $\mathbf{V} = \mathbf{M}_{L_1} W^V$, where $W^Q, W^K, W^V \in \mathbb{R}^{d \times d}$ are learnable projection matrices. The module concurrently computes two outputs: the memory-enhanced representation $\mathbf{H}_{\text{mem}}$ and the raw cross-attention scores $\alpha_{\text{cur}}$:

$$\mathbf{H}_{\text{mem}}, \alpha_{\text{cur}} = \mathcal{A}_{\text{cross}}(\mathbf{H}_{\text{cur}}, \mathbf{M}_{L_1}). \tag{1}$$

The representation $\mathbf{H}_{\mathrm{mem}}$ is thereby enriched with context from historical examples stored in the active $\mathrm{L}_1$ memory, while the attention scores $\alpha_{\mathrm{cur}} \in \mathbb{R}^{N_h \times L \times |\mathcal{M}_{\mathrm{L}_1}|}$, where $N_h$ is the number of attention heads, are used as a relevance signal for the subsequent memory update process. At inference, the same cross-attention mechanism is applied, and the query is formed only from the test input. The model encodes the partial sequence through the frozen backbone LLM, projects the final-token hidden state, and uses this vector to search over the memory bank (Sec. 3.2). This produces a memory-enhanced representation based solely on semantic similarity between the input and stored examples. Retrieval is entirely input-driven and permutation-invariant, allowing the model to integrate the most relevant memory entries during test-time adaptation.

**Memory Update.** To ensure the memory bank remains constant-sized and evolves over time, we employ a dynamic update mechanism that operates periodically every $I$ training steps. The process is driven by the attention scores $\alpha_{\mathrm{cur}}$ from $\mathcal{A}_{\mathrm{cross}}$ as defined above. For each training input, these scores are aggregated and normalized across attention heads and the input length $L$ to compute a single relevance score for every (entry) vector $\mathbf{m} \in \mathcal{M}_{\mathrm{L}_1}$: $s(\mathbf{m}) = \frac{1}{N_h L} \sum_{h=1}^{N_h} \sum_{i=1}^{L} \alpha_{\mathrm{cur}}^{(h,i,j)}$. This score is accumulated over the $I$-step interval to produce a long-term utility estimate, $S(\mathbf{m}) = \frac{1}{I}$, for each $\mathrm{L}_1$ vector. Note that the scores for vectors in the $\mathrm{L}_2$ Memory remain static during this period, retaining their last known utility value. At the end of the interval, a global update is triggered. First, all vectors in both memory pools, $\mathcal{M}_{\mathrm{L}_1} \cup \mathcal{M}_{\mathrm{L}_2}$, are ranked based on their utility scores $S$ so far. A fraction $\eta \in (0, 1)$ of the lowest-scored vectors are permanently pruned. The resulting empty slots are then replenished with an equal number of new candidate vectors generated from the most recent training phase. Finally, this updated and replenished pool of $|\mathcal{M}_{\mathrm{L}_1}| + |\mathcal{M}_{\mathrm{L}_2}|$ vectors is going to be re-ranked based on utility. The top-$|\mathcal{M}_{\mathrm{L}_1}|$ vectors are designated as the new $\mathrm{L}_1$ memory for the next training interval, ensuring it always contains the most salient examples for active interaction. The remaining vectors constitute the new $\mathrm{L}_2$ memory. This strategy maintains a clear hierarchy where the $\mathrm{L}_2$ memory serves as a robust long-term reservoir, while the $\mathrm{L}_1$ memory functions as the dynamic and compact working set. Importantly, DYNMEM does not depend on the order of memory entries: the cross-attention read-in mechanism treats the memory bank as a permutation-invariant collection. Reordering or refreshing memory vectors requires no parameter relearning, and no positional information about the memory bank is ever used by the model. See Alg. 1 for details.

**Adaptive Fusion via Learned Gating.** After obtaining the original $\mathbf{H}_{\mathrm{cur}}$ and memory-enhanced $\mathbf{H}_{\mathrm{mem}}$ representations, a learnable gating mechanism is used to fuse them. This allows the model to control the intensity of information injection or reading from the memory. Based on the input, we compute an input-aware gating coefficient $\gamma = \sigma(\mathbf{H}_{\mathrm{cur}} \mathbf{W}_g + \mathbf{b}_g)$, where $\mathbf{W}_g$ and $\mathbf{b}_g$ are the learnable parameters of the gating layer and $\sigma(\cdot)$ is the sigmoid function. The final fused representation is then passed to the final prediction layer as

$$\mathbf{H}_{\mathrm{fuse}} = (1 - \gamma) \odot \mathbf{H}_{\mathrm{cur}} + \gamma \odot \mathbf{H}_{\mathrm{mem}}. \tag{2}$$

The learnable nature of this gate allows DYNMEM to adaptively balance its reliance on model internal knowledge versus accumulated external knowledge.

### 3.2 DYNAMIC MEMORY RETRIEVAL AT INFERENCE

In addition to the dynamic interaction as described above for training, we can adopt another inference-time procedure by using more extensive dynamic query-specific knowledge retrieval. In other words, unlike the training process, which interacts with a fixed $\mathrm{L}_1$ memory to learn general patterns of example relevance, the inference process performs a global search over $\mathrm{L}_1\&\mathrm{L}_2$ to find the most pertinent context for each individual test example. For each incoming test example, we first generate a query vector $\mathbf{Q}_{\mathrm{test}}$, using the same feature extraction process as for the memory entries themselves. We then perform an efficient similarity search (*e.g.,* maximum inner-product search) to approximate the attention used in training against all vectors in the unified memory pool, $\mathcal{M}_{\mathrm{L}_1} \cup \mathcal{M}_{\mathrm{L}_2}$. This step dynamically assesses the relevance of every stored memory candidate with respect to the current input. The top-$|\mathcal{M}_{\mathrm{L}_1}|$ highest-scoring memory vectors are selected to form a sample-specific memory set, denoted as $\mathcal{M}_{\mathrm{retrieved}}$. This retrieved set is then used as context for $\mathcal{A}_{\mathrm{cross}}$, and subsequently integrated via the gated fusion mechanism to produce the final prediction. This two-stage design is critical for both efficiency and scalability 4.3. It allows DYNMEM to maintain a much larger long-term knowledge reservoir in its $\mathrm{L}_2$ memory without incurring a proportional computational cost at inference time. The model's forward pass only ever processes a small,

fixed-size set of $K$ relevant exemplars, decoupling the size of the knowledge base from the cost of prediction. Hence, the retrieval process at inference does not involve pruning of the memory bank.

---

**Algorithm 1** Memory Update of DYNMEM

---

**Require:** dataset $\mathcal{D}$; model $f_\theta$; memory bank $\mathcal{M}_{L_1}, \mathcal{M}_{L_2}$; update interval $I$; pruning ratio $\eta$
  **Training:**
  Initialize $\mathcal{M}_{L_1}, \mathcal{M}_{L_2} \leftarrow \emptyset$; set $S(\mathbf{m}) \leftarrow 0$ for all $\mathbf{m} \in \mathcal{M}_{L_1}$
                              $\triangleright S(\mathbf{m})$ stores the long-term utility score aggregated from attention weights
  **for** each training step $t = 1, 2, \ldots$ over the data stream **do**
      Sample batch $\{(x, y)\}$
      Encode example to candidate memory vector: $\mathbf{m}_{\text{new}} \leftarrow g\big(f_\theta(x, y)\big)$
                                  $\triangleright g(\cdot)$: projection of final-layer last-token state
      Obtain current hidden states $\mathbf{H}_{\text{cur}}$ from $f_\theta$
      $(\mathbf{H}_{\text{mem}}, \alpha_{\text{cur}}) \leftarrow \mathcal{A}_{\text{cross}}(\mathbf{H}_{\text{cur}}, \mathbf{M}_{L_1})$
      Fuse representations via gate: $\mathbf{H} \leftarrow (1 - \gamma)\mathbf{H}_{\text{cur}} + \gamma \mathbf{H}_{\text{mem}}$
      Compute loss $\mathcal{L}$ on $\mathbf{H}$ and update trainable parameters of DYNMEM
      Update utility $S(\mathbf{m})$ for each $\mathbf{m} \in \mathcal{M}_{L_1}$ using $\alpha_{\text{cur}}$
      Add $\mathbf{m}_{\text{new}}$ to temporary candidate pool $\mathcal{B}$
      **if** $t \bmod I = 0$ **then**
          Rank all entries in $\mathcal{M}_{L_1} \cup \mathcal{M}_{L_2}$ by $S(\cdot)$
          Prune the bottom $\eta$ fraction
          Fill freed slots with top candidates from $\mathcal{B}$; clear $\mathcal{B}$
          Re-rank the refreshed pool by $S(\cdot)$
          Assign top $|\mathcal{M}_{L_1}|$ entries to new $\mathcal{M}_{L_1}$; remainder to $\mathcal{M}_{L_2}$
          Reset $S(\mathbf{m}) \leftarrow 0$ for all $\mathbf{m} \in \mathcal{M}_{L_1}$
      **end if**
  **end for**

  **Inference:**
  Encode input $x$: $\mathbf{H}_{\text{cur}} \leftarrow f_\theta(x)$
  Compute cross-attention over memory: $(\mathbf{H}_{\text{mem}}, \alpha_{\text{cur}}) \leftarrow \mathcal{A}_{\text{cross}}(\mathbf{H}_{\text{cur}}, \mathcal{M}_{L_1} \cup \mathcal{M}_{L_2})$
  **Refresh working memory:** update $\mathcal{M}_{L_1}$ using the relevance scores $\alpha_{\text{cur}}$
  Fuse representations: $\mathbf{H} \leftarrow (1 - \gamma)\mathbf{H}_{\text{cur}} + \gamma \mathbf{H}_{\text{mem}}$
  Generate prediction $\hat{y}$ from $\mathbf{H}$

---

# 4 EXPERIMENTS

We conduct a comprehensive empirical evaluation to substantiate DYNMEM's capability as a unified framework excelling in both Continual and Simultaneous Adaptation. Our central hypothesis is that our bi-level memory architecture naturally addresses the principal challenges of each paradigm. For *Continual Adaptation*, the memory acts as an explicit knowledge reservoir to preserve past experiences, thereby mitigating catastrophic forgetting. For *Simultaneous Adaptation*, it serves as a powerful channel for instance-based knowledge sharing, enhancing transfer across tasks. Accordingly, our evaluation is structured around these two core settings, assessing performance in continual learning streams as well as in standard *Single-task Tuning* and *Multi-task Integration*. Across this diverse suite of benchmarks, we demonstrate that DYNMEM consistently surpasses specialized baselines, validating its efficacy as a truly versatile adaptation solution.

## 4.1 CONTINUAL ADAPTATION

### 4.1.1 EXPERIMENTAL SETUP

**Task Streams.** We construct two challenging task streams to evaluate performance under different conditions of semantic shift: a) STRUCTURED STREAM. This stream contains 4 structured knowledge reasoning datasets: Spider (Yu et al., 2018) for text-to-SQL, ComplexWebQuestions (Talmor & Berant, 2018) for text-to-SPARQL, GrailQA (Gu et al., 2021) for S-expression generation, and MTOP (Li et al., 2021) for semantic parsing in dialogue systems. These tasks focus on structured

language generation, testing the model's capability to retain reasoning skills on structured data. b) MIXED STREAM. To simulate more significant domain shifts, we augment the Structured Stream with four diverse commonsense reasoning datasets: BoolQ (Clark et al., 2019), PIQA (Bisk et al., 2020), Arc-Easy (Clark et al., 2018), WinoGrande (Sakaguchi et al., 2021). This stream tests the model's robustness to maintain distinct knowledge bases. For each task in both streams, we create a standardized split by randomly sampling 1,000 examples for training and 300 examples for testing.

**Evaluation Metrics.** We adopt standard metrics from continual learning literature (Chen et al., 2023). Let $A_{k,i}$ be the accuracy on the test set of task $\mathcal{T}_i$ after the model has finished training on task $\mathcal{T}_k$. a) *Average Accuracy*: $\text{Acc} = \frac{1}{k} \sum_{i=1}^{k} A_{k,i}$ reflects the average accuracy on all tasks seen so far after training on task $\mathcal{T}_k$; b) *Backward Transfer*: $\text{BWT} = \frac{1}{k-1} \sum_{i=1}^{k-1} (A_{k,i} - A_{i,i})$ measures the degree of model forgetting; c) *Forward Transfer*: $\text{FWT} = \frac{1}{k-1} \sum_{i=2}^{k} (A_{i,i} - Ai, 0)$ evaluates model's ability to transfer previously-learned knowledge to new tasks.

**Compared Methods.** We compare DYNMEM against a comprehensive set of baselines: a) *Fine-tuning:* A vanilla fine-tuning approach where the model is updated on each task sequentially, which usually exhibits severe forgetting. b) *PEFT Methods:* This family of methods leverages parameter-efficient tuning techniques by allocating a separate, small set of trainable parameters for each task. We include Continual Prompt Tuning (CPT)(Zhu et al., 2022) and C3 (Chen et al., 2023) as representatives. c) *Rehearsal-based Methods:* We compare against EMAR (Han et al., 2020), a baseline that stores a buffer of raw examples from past tasks and rehearses them when learning a new task.

**Backbone Models.** We conduct experiments on two powerful open-source large language models: Llama-3-8B (Dubey et al., 2024) and Qwen-3-8B (Yang et al., 2025).

**Implementation Details.** For DYNMEM, we set the $L_1$ memory size to $|\mathcal{M}_{L_1}| = 100$ and the $L_2$ to $|\mathcal{M}_{L_2}| = 1000$. The memory is updated periodically with an interval of $I = 500$ training steps.

### 4.1.2 EXPERIEMNTAL RESULTS

Table 1: Continual Adaptation on the Structured and Mixed streams. ORACLE serves as an upper-bound performance, representing a model trained jointly on the union of all task datasets; EMAR utilizes a buffer of 10 examples per past task. Each cell shows mean$_{\text{std}}$.

| Backbone | Method | Structured | | | Mixed | | |
|---|---|---|---|---|---|---|---|
| | | Acc | BWT | FWT | Acc | BWT | FWT |
| Llama-3-8B | FINE-TUNING | $22.3_{3.4}$ | $-26.3_{3.4}$ | $2.3_{1.2}$ | $40.3_{5.4}$ | $-19.1_{3.8}$ | $3.4_{1.3}$ |
| | EMAR | $35.3_{2.3}$ | $-14.2_{1.3}$ | $2.3_{0.4}$ | $49.5_{2.0}$ | $-15.3_{2.3}$ | $2.9_{0.7}$ |
| | CPT | $10.3_{4.5}$ | - | $0.7_{0.3}$ | $23.6_{3.1}$ | - | $2.3_{0.4}$ |
| | C3 | $37.2_{3.1}$ | - | $3.7_{1.4}$ | $51.6_{3.9}$ | - | $4.1_{1.6}$ |
| | DYNMEM | $41.2_{4.0}$ | $-12.3_{2.1}$ | $5.6_{1.5}$ | $57.2_{3.6}$ | $-13.4_{1.2}$ | $7.6_{1.8}$ |
| | ORACLE | $56.2_{0.9}$ | $6.5_{1.9}$ | $6.7_{2.4}$ | $61.4_{1.5}$ | $8.7_{2.3}$ | $8.1_{0.6}$ |
| Qwen-3-8B | FINE-TUNING | $25.4_{3.2}$ | $-19.8_{4.3}$ | $2.4_{0.6}$ | $45.2_{6.7}$ | $-18.2_{2.5}$ | $4.1_{0.6}$ |
| | EMAR | $37.1_{3.4}$ | $-13.1_{4.2}$ | $3.1_{0.8}$ | $52.0_{3.1}$ | $-18.7_{0.9}$ | $3.0_{0.9}$ |
| | CPT | $13.4_{1.7}$ | - | $1.3_{0.5}$ | $24.2_{2.8}$ | - | $3.1_{0.7}$ |
| | C3 | $38.2_{3.7}$ | - | $3.2_{0.7}$ | $54.6_{3.8}$ | - | $3.5_{0.8}$ |
| | DYNMEM | $43.9_{2.3}$ | $-10.3_{1.7}$ | $7.1_{3.6}$ | $60.1_{0.8}$ | $-11.9_{2.8}$ | $8.0_{1.2}$ |
| | ORACLE | $59.1_{3.4}$ | $10.1_{3.5}$ | $8.2_{1.2}$ | $62.7_{2.0}$ | $9.1_{0.7}$ | $7.9_{1.3}$ |

**Overall Results.** The results in Table 1 confirm that DYNMEM establishes a new state-of-the-art in continual learning. On MIXED STREAM using Llama-3-8B, DYNMEM achieves an average accuracy of 57.2, substantially outperforming the strongest baseline, C3, by 5.6 points. This superior performance is a direct result of our memory-centric design, which excels at both mitigating forgetting and accumulating new knowledge. DYNMEM achieves the highest Backward Transfer, demonstrating that the bi-level memory acts as a robust knowledge reservoir where inference-time retrieval successfully compensates for parametric drift. DYNMEM also achieves the highest Forward Transfer, nearly doubling that of the next best method, showing the memory does not merely preserve old knowledge but actively facilitates the learning of new tasks by providing relevant, instance-based

context to accelerate adaptation. Unlike PEFT methods that prevent forgetting at the cost of limited knowledge sharing (lower FWT), DYNMEM's unified architecture enables both strong knowledge preservation and positive transfer, offering a more effective and holistic solution to the continual learning problem.

**Performance till Seen Tasks.** To provide a more granular view of the learning dynamics, we plot the average accuracy on all previously seen tasks as the model progresses through both the STRUCTURED and MIXED streams in Figure 2. This visualization vividly illustrates the models' ability to accumulate and retain knowledge over time. As expected, standard Fine-tuning suffers a precipitous decline in average accuracy, clearly demonstrating catastrophic forgetting. While other methods like EMAR and C3 offer partial mitigation, they ultimately succumb to a steady degradation of knowledge as more tasks are introduced, with their performance curves showing a clear downward trend.

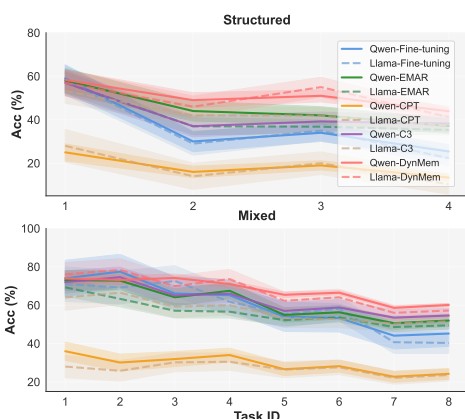

DYNMEM exhibits remarkable stability and a strong capacity for knowledge accumulation across both streams and backbone models. Its performance trajectory remains high and relatively flat, showing only a minor initial drop before stabilizing. This pro-

Figure 2: Acc on all seen tasks as the model trains sequentially on two streams.

vides compelling evidence that our dynamic bi-level memory system successfully decouples knowledge preservation from parametric adaptation: the LLM learns the new task, while the memory update and inference-time retrieval mechanisms successfully preserve and leverage knowledge from the past, enabling robust and stable knowledge accumulation over the entire task sequence.

## 4.2 SIMULTANEOUS ADAPTATION

### 4.2.1 EXPERIMENTAL SETUP

**Datasets.** We evaluate simultaneous adaptation performance on a comprehensive suite of widely-used benchmarks covering commonsense reasoning. Specifically, we use eight commonsense reasoning datasets: BoolQ (Clark et al., 2019), PIQA (Bisk et al., 2020), SocialIQa (SIQA) (Sap et al., 2019), HellaSwag (HellaS.) (Zellers et al., 2019), WinoGrande (WinoG.) (Sakaguchi et al., 2021), ARC-Easy (ARC-e) (Clark et al., 2018), ARC-Challenge (ARC-c) (Clark et al., 2018), and Open-BookQA (OBQA) (Mihaylov et al., 2018). We also include GSM8K (Cobbe et al., 2021) for mathematical reasoning.

**Evaluation Protocol.** We evaluate all methods under two primary simultaneous settings: a) *Single-task Tuning*. The model is fine-tuned and evaluated on each dataset separately to measure its performance on corresponding tasks. b) *Multi-task Tuning*. A single model is jointly trained on the combined training sets of all eight commonsense reasoning datasets and is subsequently evaluated on the test set of each individual task to measure knowledge integration.

**Compared Methods.** We benchmark DYNMEM against a comprehensive set of state-of-the-art PEFT methods to ensure a rigorous comparison: LoRA (Hu et al., 2022), NoRA (Lin et al., 2024), LoKr (Yeh et al., 2024), DoRA (Liu et al., 2024), AdaLoRA (Zhang et al., 2023), MixLoRA (Li et al., 2024), and DenseLoRA (Mu et al., 2025).

**Backbone Models.** To ensure our findings are robust and generalizable, we conduct all experiments on two powerful open-source large language models: Llama-3-8B and Qwen-3-8B.

The results for Single-task Tuning, presented in Table 2, reveal the superiority of DYNMEM as a fine-tuning framework. Across both backbone models, DYNMEM consistently establishes a new state-of-the-art, outperforming all compared PEFT baselines.

Focusing on the Llama-3-8B results, DYNMEM achieves an average score of 86.8 on the commonsense reasoning benchmarks, a salient improvement of 1.9 points over the strongest baseline,

Table 2: Experiments on Single-task Tuning. See B for parameter calculation details.

| Method | Param (%) | Commonsense Reasoning | | | | | | | | | Math |
| | | BoolQ | PIQA | SIQA | HellaS. | WinoG. | ARC-e | ARC-c | OBQA | Avg. | GSM8K |
| Llama-3-8B | | | | | | | | | | | |
| LoRA | 0.35 | 72.3 | 86.7 | 79.3 | 93.5 | 84.8 | 87.7 | 75.7 | 82.8 | 82.9 | 57.2 |
| DoRA | 0.36 | 73.3 | 89.1 | 79.9 | 95.9 | 84.7 | 89.8 | 79.5 | 86.9 | 84.9 | 58.1 |
| AdaLoRA | 0.35 | 75.2 | 88.2 | 79.2 | 76.2 | 85.2 | 89.9 | 78.2 | 85.0 | 82.1 | 52.1 |
| MixLoRA | 2.60 | 75.0 | 87.6 | 78.8 | 93.3 | 82.1 | 86.5 | 79.9 | 84.8 | 83.5 | 55.6 |
| DynMem | 0.18 | 76.5 | 90.3 | 81.9 | 96.6 | 86.0 | 93.3 | 80.7 | 89.3 | 86.8 | 59.6 |
| Qwen-3-8B | | | | | | | | | | | |
| LoRA | 0.35 | 75.6 | 91.0 | 81.5 | 92.7 | 88.6 | 95.6 | 89.7 | 92.8 | 88.4 | 64.5 |
| DoRA | 0.36 | 76.1 | 89.9 | 82.1 | 93.7 | 87.9 | 96.8 | 89.3 | 93.1 | 88.6 | 63.1 |
| DynMem | 0.18 | 76.5 | 91.2 | 90.7 | 95.6 | 88.4 | 96.2 | 91.5 | 91.3 | 90.2 | 65.6 |

DoRA (84.9). This trend also holds for mathematical reasoning, where DynMem achieves 59.6 on GSM8K, again surpassing all baselines. Crucially, DynMem achieves these state-of-the-art results while being significantly more parameter-efficient. With only 0.18% trainable parameters with regard to the backbone LLM, it uses approximately half the parameters of LoRA/DoRA and an order of magnitude fewer than methods like MixLoRA.

This pattern of superior performance and efficiency is not limited to a single model architecture. As the results for the Qwen-3-8B backbone confirm, DynMem consistently outperforms the baseline methods, demonstrating that its advantages are generalizable across different foundational models.

These results highlight a key advantage of our approach. Unlike purely parametric methods that compress all task knowledge into a small set of adapter weights, DynMem leverages its memory to store and retrieve the most salient examples from the training data. This instance-based conditioning provides powerful, explicit context at inference time, leading to more robust intra-task generalization and ultimately higher accuracy. Therefore, even in this fundamental adaptation setting, DynMem proves to be a more effective and efficient fine-tuning solution.

### 4.2.2 MULTI-TASK INTEGRATION

Table 3: Experiments on Multi-task Integration.

| Method | Param (%) | Commonsense Reasoning | | | | | | | | |
| | | BoolQ | PIQA | SIQA | HellaS. | WinoG. | ARC-e | ARC-c | OBQA | Avg. |
| Llama-3-8B | | | | | | | | | | |
| LoRA | 0.35 | 72.3 | 83.7 | 78.1 | 91.6 | 82.8 | 84.9 | 72.4 | 81.2 | 80.9 |
| NoRA | 0.10 | 73.3 | 86.4 | 79.1 | 94.1 | 84.3 | 88.2 | 77.5 | 85.0 | 83.5 |
| LoKr | 0.01 | 65.1 | 81.6 | 78.7 | 92.0 | 82.1 | 89.2 | 76.7 | 80.9 | 80.8 |
| DoRA | 0.35 | 71.8 | 86.1 | 79.4 | 94.0 | 85.1 | 88.0 | 77.4 | 87.2 | 83.6 |
| AdaLoRA | 0.35 | 75.1 | 86.4 | 76.7 | 75.4 | 83.3 | 90.4 | 79.1 | 85.0 | 81.4 |
| MoSLoRA | 0.36 | 74.6 | 89.7 | 81.0 | 95.0 | 85.8 | 90.5 | 81.5 | 86.8 | 85.6 |
| DenseLoRA | 0.06 | 74.1 | 88.9 | 80.3 | 95.0 | 87.0 | 90.0 | 79.2 | 85.6 | 85.0 |
| DynMem | 0.18 | 76.5 | 89.3 | 81.1 | 95.4 | 85.7 | 93.3 | 80.7 | 88.3 | 86.3 |
| Qwen-3-8B | | | | | | | | | | |
| LoRA | 0.35 | 73.2 | 88.1 | 80.4 | 90.0 | 86.1 | 93.0 | 87.9 | 91.8 | 86.3 |
| DoRA | 0.36 | 75.0 | 88.1 | 79.9 | 91.2 | 87.0 | 94.9 | 88.8 | 92.9 | 87.2 |
| DynMem | 0.18 | 74.5 | 90.0 | 90.7 | 94.3 | 87.9 | 96.0 | 90.3 | 90.3 | 89.3 |

The results for Multi-task Tuning (MT), presented in Table 3, highlight DynMem 's exceptional capability for knowledge integration and transfer across a diverse set of tasks.

On the **Llama-3-8B** backbone, DynMem achieves a new state-of-the-art with an average score of 86.4 across all eight commonsense reasoning datasets. This represents an improvement of 0.8 points over the strongest PEFT baseline, MoSLoRA (85.6), and demonstrates superior performance on nearly every individual task. Notably, this superior performance is achieved with significantly higher parameter efficiency; DynMem uses only a smaller amount of trainable parameters.

This strong performance in a multi-task setting validates the core design of our framework. While standard PEFT methods rely on implicit knowledge transfer through a shared set of parameters,

DYNMEM introduces a powerful channel for explicit, instance-based knowledge sharing. During training, the memory update process populates $L_1$ and $L_2$ Memory with the most salient examples from co-trained tasks. At inference, the retrieval mechanism can fetch a transferable relevant example. This targeted, cross-task retrieval allows DYNMEM to leverage inter-task synergies effectively than purely parametric approaches, leading to a more capable and integrated multi-task model.

## 4.3 ABLATION STUDY

To validate the contribution of each core component of our design, we conduct an ablation study, the results of which are presented in Table 4. We evaluate several variants of DYNMEM by removing one key mechanism at a time: a) *w/o $L_2$ Memory* tests the necessity of the long-term reservoir by using only a single $L_1$ memory; b) *w/o Inference Retrieval* assesses the benefit of dynamic, query-specific retrieval by using the static $L_1$ memory for inference; c) *w/o Gated Fusion* replaces the learned gate with a static value $0.5$ to measure the importance of adaptive integration; and d)*w/o Attention Ranking* replaces our utility-based update with a simple First In First Out strategy to test the efficiency memory management strategy. The results confirm that all components are critical, as removing any of them substantially degrades performance across all settings. The degradation is most severe for *w/o Gated Fusion*, confirming the necessity of adaptively controlling information flow from the memory. Disabling the $L_2$ *Memory*, Inference Retrieval, and *Attention Ranking* also significantly impairs performance, validating the respective benefits of a large knowledge reservoir, query-specific context, and intelligent memory management.

Table 4: DYNMEM Component Ablation Study.

| Method | Continual | | | | | | Simultaneous | | |
| --- | --- | --- | --- | --- | --- | --- | --- | --- | --- |
| | Structured | | | Mixed | | | Single-task | | Multi-task |
| | Acc | BWT | FWT | Acc | BWT | FWT | Com.S. | Math | Com.S. |
| **Llama-3-8B** | | | | | | | | | |
| DYNMEM | 41.2 | -12.3 | 5.6 | 57.2 | -13.4 | 7.6 | 86.8 | 59.6 | 86.4 |
| w/o $L_2$ Memory | 36.1 | -18.9 | 4.1 | 52.0 | -18.5 | 6.2 | 84.5 | 57.1 | 84.1 |
| w/o Inference Retrieval | 37.8 | -17.2 | 4.9 | 53.8 | -16.9 | 6.8 | 85.0 | 57.9 | 84.9 |
| w/o Gated Fusion | 13.5 | -45.4 | 1.0 | 24.6 | -35.1 | 2.2 | 56.2 | 32.7 | 43.5 |
| w/o Attention Curation | 33.4 | -21.5 | 3.5 | 48.5 | -22.1 | 5.4 | 82.3 | 54.2 | 81.9 |
| **Qwen-3-8B** | | | | | | | | | |
| DYNMEM | 43.9 | -10.3 | 7.1 | 60.1 | -11.9 | 8.0 | 90.2 | 65.6 | 89.3 |
| w/o $L_2$ Memory | 38.2 | -16.5 | 5.5 | 54.7 | -17.3 | 6.5 | 87.9 | 62.8 | 86.8 |
| w/o Inference Retrieval | 40.1 | -14.9 | 6.1 | 56.2 | -15.6 | 7.2 | 88.5 | 63.5 | 87.7 |
| w/o Gated Fusion | 16.0 | -41.8 | 2.0 | 19.1 | -38.6 | 2.1 | 61.6 | 37.0 | 48.7 |
| w/o Attention Curation | 35.5 | -19.8 | 4.6 | 50.8 | -20.5 | 5.9 | 85.1 | 60.1 | 84.4 |

To assess DYNMEM's scalability, we vary the $L_1$ and $L_2$ memory capacities and find that performance monotonically increases with the memory budget across all Continual, Single-task, and Multi-task settings (Figure 3). This scalability is a direct result of our architecture: a larger $L_2$ memory provides a more comprehensive reservoir for inference-time retrieval, while a larger $L_1$ memory offers a richer context for training-time interaction and curation. This confirms that DYNMEM effectively capitalizes on available resources, offering a clear and predictable trade-off between performance and memory overhead, making it viable for a wide range of computational budgets.

## 5 RELATED WORK

**Task Adaptation of Language Models.** The dominant paradigm for adapting large language models is Parameter-Efficient Fine-Tuning (PEFT). The seminal LoRA method (Hu et al., 2022) and its successors (Zhang et al., 2023; Liu et al., 2024) drastically reduce computational costs by fine-tuning a small set of auxiliary parameters while keeping the backbone model frozen. While highly effective for single-task (Li et al., 2024) or static multi-task (Lin et al., 2024; Wu et al., 2024; Mu et al., 2025) scenarios, these methods are not inherently suited for continual learning. The primary challenge is that they sequentially either overwrite the adapter weights, causing catastrophic forgetting, or require storing an ever-growing set of per-task modules, which introduces significant parameter overhead (Zhu et al., 2022; Chen et al., 2023). We design a new dynamic memory module that is not only parameter-efficient but also effective in both continual and simultaneous adaptation scenarios.

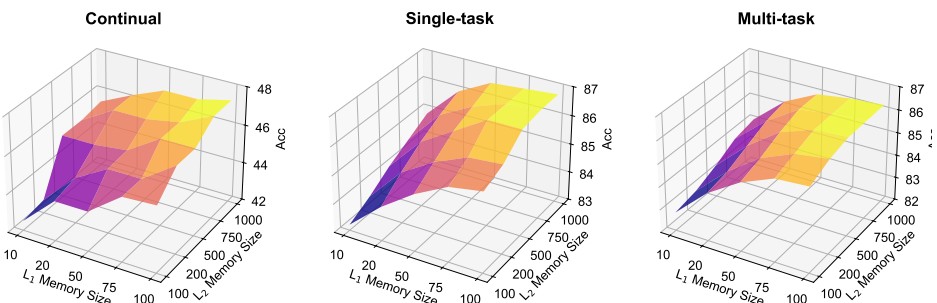

Figure 3: Performance analysis of DYNMEM on Llama-3-8B as a function of $L_1$ and $L_2$ memory sizes. The plots show final accuracy for Continual (Structured Stream), Single-task Tuning (Commonsense Avg.), and Multi-task Tuning (Commonsense Avg.). Performance consistently increases with larger memory capacities across all three paradigms, demonstrating excellent scalability.

**Memory-augmented Models.** Classified by Yang et al. (2024), popular memory-augmented model architectures include: a) Retrieval-augmented methods (Wu et al., 2022b), which extend the effective context length by caching and retrieving past hidden states, enabling models to handle sequences beyond their native context window; b) Memory-augmented transformers like Memformer (Wu et al., 2022a; Kang et al., 2025) introduce structured memory slots directly into the self-attention mechanism, improving long-range sequence modeling; c) Parameter-as-memory approaches reinterpret model weights as implicit knowledge storage, recent works (Mitchell et al., 2022; Wang et al., 2024) manipulate or reorganize these knowledge neurons to update factual content at scale. Inspired by these methods, we design a novel bi-level dynamic memory to effectively maintain long-term information for positive knowledge sharing across tasks learned either simultaneously or sequentially.

## 6 CONCLUSION

In this work, we address a critical gap in the adaptivity of pre-trained LLMs by introducing DYN-MEM, a unified memory-augmented method that supports both continual and simultaneous learning. The core part of our method is a bi-level dynamic memory system based on attention-based retrieval and pruning. DYNMEM enables efficient, example-level knowledge retention and dynamic integration without modifying the backbone LLM. Our design not only mitigates catastrophic forgetting in sequential adaptation but also enhances generalization in multi-task settings, all while maintaining a compact memory footprint. Extensive experiments across various benchmarks show that DYNMEM consistently outperforms baselines, establishing a new state-of-the-art for versatile LLM adaptation.

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

## A  THE USE OF LARGE LANGUAGE MODELS

Large language models were used as writing assistants to aid or polish the text in this manuscript. Their functions were limited to improving clarity, grammar, and professional tone. All scientific contributions, including the core methodology, experimental design, and conclusions, are the original work of the authors, who retain full responsibility for the paper's content.

## B  PARAMETER CALCULATION DETAILS

This section provides a detailed breakdown of how the trainable parameter percentages reported in our main results (e.g., Table 2) are calculated. The percentage reflects the ratio of trainable parameters to the total number of parameters in the frozen backbone model.

Let $P_{\text{trainable}}$ be the number of parameters that are updated during fine-tuning, and let $P_{\text{total}}$ be the total number of parameters in the backbone model (e.g., Llama-3-8B). The reported percentage is calculated as:

$$\text{Param}(\%) = \left( \frac{P_{\text{trainable}}}{P_{\text{total}}} \right) \times 100 \tag{3}$$

The composition of $P_{\text{trainable}}$ differs between the baseline methods and our proposed DYNMEM framework.

**PEFT Baselines (e.g., LoRA, DoRA).**  For methods such as LoRA and its variants, $P_{\text{trainable}}$ exclusively comprises the parameters of the injected low-rank adaptation matrices and any other small, method-specific modules (e.g., the magnitude vectors in DoRA). The vast majority of the original model weights remain frozen. We set $\text{lora\_rank} = 16, \text{lora\_alpha} = 32$ for these baselines.

**DYNMEM.**  For our DYNMEM framework, $P_{\text{trainable}}$ consists of the parameters from our lightweight, newly introduced memory interaction modules, which are trained jointly with the backbone model. Specifically, these include:

- The cross-attention projection matrices: $W^Q, W^K, W^V \in \mathbb{R}^{d \times d}$.
- The gated fusion layer parameters: $\mathbf{W}_g \in \mathbb{R}^{d \times d}$ and $\mathbf{b}_g \in \mathbb{R}^d$.

In our implementation, both the backbone model $f_\theta$ and these interaction modules are fine-tuned. However, as the results demonstrate, the total number of trainable parameters in DYNMEM's modules is significantly smaller than that of many PEFT baselines, highlighting its parameter efficiency. For DYNMEM, we use $lora\_rank = 8, lora\_alpha = 16$ to achieve a more efficient training.

## C  TASK ORDERS OF CONTINUAL ADAPTATION

To ensure the robustness of our findings in the asynchronous adaptation setting and to mitigate any potential bias resulting from a single, arbitrary task sequence, we conducted all continual learning experiments across three different permutations for both the STRUCTURED STREAM and the MIXED STREAM. The final results reported in the main paper represent the mean and standard deviation across these three runs. The specific task orders for each stream are detailed below.

### C.1  STRUCTURED STREAM

This stream consists of four structured knowledge reasoning datasets. The three specific task orders used in our experiments are as follows:

- **Order 1:** GrailQA → MTOP → Spider → ComplexWebQuestions
- **Order 2:** ComplexWebQuestions → Spider → MTOP → GrailQA
- **Order 3:** MTOP → GrailQA → ComplexWebQuestions → Spider

## C.2 MIXED STREAM

This stream interleaves the four structured datasets with four commonsense reasoning datasets to introduce more significant domain shifts. The three specific task orders used are as follows:

- **Order 1:** GrailQA → BoolQ → MTOP → PIQA → Spider → ARC-e → ComplexWebQuestions → WinoGrande
- **Order 2:** ComplexWebQuestions → WinoGrande → Spider → ARC-e → MTOP → PIQA → GrailQA → BoolQ
- **Order 3:** MTOP → PIQA → GrailQA → BoolQ → ComplexWebQuestions → WinoGrande → Spider → ARC-e

