# OpenReview forum: "Coupling Attention and Memory: A Dynamic Memory Module for Efficient Adapation with Pretrained LLMs"
_ICLR.cc/2026/Conference — Submitted to ICLR 2026_

### Official Review · Reviewer_BULr · 2025-10-29

**Soundness:** 2
**Presentation:** 2
**Contribution:** 2
**Rating:** 2
**Confidence:** 4

**Summary:**

This paper proposes a novel framework, DynMem, designed to address both the interference issue in multi-task learning and the catastrophic forgetting problem in continual learning. This framework introduces a bi-level memory module (i.e., L1 for short-term, L2 for long-term) that continuously re-ranks its entries based on attention scores. Through this dynamic memory mechanism, DynMem enables parameter-efficient adaptation and inference.

**Strengths:**

- This paper is easy to follow.

- Rather than focusing on a single challenge, the authors design a unified and well-structured module that jointly addresses both catastrophic forgetting and interference issue.

- The proposed framework is highly parameter-efficient, as it freezes the backbone LLM and trains only a very small set of additional parameters.

- The experimental setup is rigorous, carefully considering domain shift through diverse evaluation streams.

**Weaknesses:**

W1. Unclear theoretical justification for some of its key design choices:
- While the bi-level design memories serves as the structural core of the framework, the current discussion focuses mainly on implementation and empirical validation. A more explicit conceptual rationale for clarifying the underlying principles that this hierarchical organization contributes to tackling the challenges of continual and multi-task learning would make the architectural choice more persuasive and theoretically grounded.

- Although the memory entries are continuously re-ranked based on attention scores, there is no strong evidence that this ranking remains relevant to the current input. It is unclear whether, at the point of significant domain shift, the stored memory might even introduce negative interference rather than helping adaptation.

- While the ablation study highlights the importance of the gated fusion mechanism, the paper does not provide analytical evidence showing that the gating coefficient effectively controls the strength of information injection as claimed.


W2. Hyperparameter Sensitivity:
- One of the core design factors in this work is the periodic memory update mechanism. However, this frequency is likely to be strongly correlated with the dataset size. For example, when training set involves a collection of smaller datasets, a faster update interval would likely be more beneficial, preventing inefficient or outdated entries from persisting in memory. Therefore, analyzing this relationship between dataset characteristics and update frequency is crucial for practical deployment, yet the paper lacks a detailed discussion or empirical study on this sensitivity.

W3. Asymmetric methodological experimental settings:
- The training dataset consists of 1,000 examples, which coincides with the sizes of the L1 and L2 memories, making the experimental setup overly favorable to DynMem. In contrast, other methods such as EMAR are evaluated under asymmetric conditions, for instance by setting the buffer size to 10.


W4. The key limitations identified about previous works — (1) the need to know which adapter to use for each task, and (2) the lack of transferability between adapters — have largely been addressed by methods based on the Mixture of Experts (MoE) framework [1,2]. You should reference these works.
The paper provides little discussion of MoE approaches and lacks a direct comparison with them. Moreover, the baselines used for continual learning tasks are mostly from 2022–2023, which makes them somewhat outdated compared to these references.
[1] Xu Owen He, Mixture of A Million Experts, 2024
[2] Jiazuo Yu, et al., Boosting Continual Learning of Vision-Language Models via Mixture-of-Experts Adapters, 2024


W5. $W$ is sensitive to the order of $M$, but since the order of memory matrix $M$ changes after each update, $W$ must be relearned from scratch every time. In contrast, adapter parameters can be reused without additional training, for a task once trained. In real-world continual scenarios with many recurring tasks, adapters might be more training-efficient.


W6. $M$ is composed of QA pairs, with each row containing one QA pair. Therefore, $M$ selectively references specific samples from the QA dataset, which can make it sample-efficient.
Adapters, on the other hand, usually capture the average distribution over the entire QA dataset, and compress them in the parameters of adapter . So it might not be sample-efficient, but might more precisely comprehend the task.
From this analysis, there comes several concerns.
- DynMem can likely achieve better performance with short training. However, with longer training, adapter-based models might perform better. It would be helpful if Table 1 included results for adapters trained over multiple epochs. (Of course, DynMem’s fast adaptation remains an advantage.)
- Using $M$ may be sensitive to the selected samples. So, when the dataset is highly diverse, a few stored samples may not adequately represent the overall distribution. Benchmarks like PIQA have relatively consistent QA formats, so this approach can work well there. However, for natural, open-ended QA tasks with more diverse distribution (such as ELI5), A few selected samples captured in $M$ would not be sufficient to cover the task.

**Questions:**

- It would be interesting to include an additional study on the effect of memory update frequency during training.

- It would be helpful to include an algorithm or example illustrating the full training and memory update process for clarity.

-  It would be beneficial to include additional experiments comparing performance when the dataset size exceeds the capacities of the L1 and L2 memories, to assess whether the proposed method remains effective under such conditions.

- Backward transfer is measured as $\sum_{k=1}^{k-1} A_{i,k}$, which means “train on $k-1$ (or <k-1) and test on $k$. This does not seem to capture forgetting, rather it measures how well the model handles unseen task. More right form seems to be $A_{k,i}$.

---

> ### Author Response · Authors · 2025-11-20
> **Re: BULr**
>
> ## Re: Weakness 1
> We appreciate the reviewer's detailed comments. Our goal in this work is primarily empirical, and we do not claim formal guarantees. That said, we agree the conceptual rationale can be made more explicit, and we will expand §3 accordingly. Intuitively, L1 serves as a small, highly plastic working set of examples that are frequently accessed and updated, while L2 acts as a more stable reservoir preserving historically useful but less recently attended examples. This hierarchy lets the model rapidly adapt in L1 without discarding globally important information preserved in L2, which is exactly what we empirically observe across continual and multi-task settings.
>
> On the concern that attention-based ranking may become misaligned after domain shifts and even introduce negative interference: in DynMem, attention scores are continuously re-estimated on new data (See §3.1) and used to periodically refresh both L1 and L2, so memory composition gradually adapts to the new domain instead of being frozen. Moreover, at inference we do not rely on stale scores alone: each test query forms its own vector and performs a fresh search over $M_{L1} \cup M_{L2}$, so retrieved entries are directly conditioned on the current input (§3.2). In the continual setting,especially the Mixed stream, the task sequence already includes substantial domain shifts (e.g., commonsense reasoning → structured semantic parsing), and DynMem maintains stable performance across these transitions, further suggesting that the ranking mechanism remains aligned even under large distributional changes. To further stress-test negative interference, we add a new experiment where, in the single-task setting, we merge all commonsense memories into a large shared bank (L1 fixed at 100, L2 expanded to 8,000) and evaluate each task with its own checkpoint, without any additional tuning. Despite the injected *foreign* examples, performance remains stable or slightly improves (please refer to our reply to Reviewer BW7u's Question 2 for more experimental details).
>
> Regarding the gating mechanism, the coefficient $\gamma$ interpolates between $H_{cur}$ and $H_{mem}$, so by construction it directly controls the strength of memory injection.  To examine whether the gating coefficient $\gamma$ meaningfully regulates memory usage, we perform a diagnostic analysis across all eight commonsense reasoning datasets using the trained single-task checkpoints. For each dataset, we take its single-task DynMem (Llama-3-8B) checkpoint and evaluate every test example under two controlled conditions: (i) the normal DynMem inference, where memory retrieval and the learned gated fusion operate as designed, and (ii) a no-memory variant, where we apply LoRA-tuned checkpoints w/o memory modules. Based on the correctness of these two predictions, we partition the test examples into three categories: (a) Memory-Helpful (incorrect without memory but correct with memory), (b) Neutral (identical predictions), and (c) Memory-Harmful (correct without memory but incorrect with memory). For each dataset and bucket, we compute the number of examples, accuracy with and without memory, and the average gating coefficient $\bar{\gamma}$. This setup, using checkpoints trained exclusively within each task, enables a clean assessment of whether the gate appropriately increases when memory is beneficial and suppresses memory when it could introduce interference.
>
>
> | Dataset    | Memory-Helpful γ | Neutral γ | Memory-Harmful γ|
> |------------|------------------|-----------|-----------------|
> | BoolQ      | 0.224            | 0.210     | 0.192           |
> | PIQA       | 0.188            | 0.185     | 0.171           |
> | SIQA       | 0.225            | 0.192     | 0.189           |
> | HellaSwag  | 0.250            | 0.243     | 0.238           |
> | WinoGrande | 0.170            | 0.193     | 0.132           |
> | ARC-e      | 0.109            | 0.112     | 0.098           |
> | ARC-c      | 0.160            | 0.162     | 0.155           |
> | OBQA       | 0.220            | 0.235     | 0.218           |
>
> The results show that the learned gating coefficient $\gamma$ varies adaptively across tasks and across example categories, rather than collapsing to a trivial constant. This variability indicates that the gate selectively regulates the contribution of memory depending on the task and instance, providing evidence that $\gamma$ indeed controls the strength of memory injection as intended.

---

> ### Author Response · Authors · 2025-11-20
> **Re: BULr Part II**
>
> ## Re: Weakness 2
> We appreciate the reviewer's comment. In our experiments, we intentionally adopt a single fixed memory-update interval across all datasets to avoid per-dataset over-tuning and to ensure that DynMem is evaluated under a generalizable, deployment-friendly configuration. To illustrate the robustness of this choice, we provide our experimental logs on GSM8K (7,473 training examples) and HellaSwag (39,905 training examples) across a wide range of update intervals $I \in \{1, 10, 20, 50, 100, 200, 500, 1000\}$. As shown in the table below, performance remains highly stable across intervals. This indicates that memory-update frequency may not have significant impact on the effectiveness of DynMem. Based on this observation, we adopt I = 500 as the unified setting throughout the paper.
>
> | Interval I | GSM8K Acc ↑ | HellaS. Acc ↑    |
> |------------|-------------|------------------|
> | 1          | 58.8        | 95.5             |
> | 10         | 59.1        | 96.0             |
> | 20         | 59.0        | 97.0             |
> | 50         | 57.9        | 96.3             |
> | 100        | 60.1        | 96.8             |
> | 200        | 59.2        | 96.4             |
> | **500**    | **59.6**    | **96.6**         |
> | 1000       | 59.7        | 96.2             |
>
>
> ## Re: Weakness 3
> We appreciate the reviewer’s concern. Our method uses a fixed-size memory bank, whereas traditional replay-based methods (e.g., EMAR) must store raw historical examples for every task, causing memory cost to grow linearly with the number of tasks. This difference explains why EMAR typically uses a small per-task buffer (e.g., 10 examples) to keep total storage manageable, while DynMem maintains a constant-size memory budget independent of task-sequence length.
>
> To provide a more balanced comparison, we add an additional experiment on the Mixed stream with matched memory scales. For DynMem, we set $|M_{L_1}| = |M_{L_2}| = 50$ (100 memory vectors total). For EMAR, we store 20 examples per task, resulting in $20 \times 8 = 160$ stored examples. This setup gives EMAR a larger total memory budget while keeping the overall scale comparable. The results for Llama-3-8B and Qwen-3-8B are shown below.
>
> | Backbone     | Method   | Acc ↑ | BWT ↑ | FWT ↑ |
> |--------------|----------|-------|-------|-------|
> | Llama-3-8B   | EMAR     | 52.0  | –13.8 | 2.9   |
> |              | DynMem   | 54.9  | –14.0 | 5.9   |
> | Qwen-3-8B    | EMAR     | 55.6  | –17.9 | 3.5   |
> |              | DynMem   | 57.7  | –14.3 | 6.9   |
>
> Even when DynMem is given a smaller total memory budget than EMAR, it continues to outperform replay-based baselines, supporting that the observed gains are not due to asymmetric memory allocation but due to the effectiveness of the proposed dual-memory mechanism.

---

> > ### Author Response · Authors · 2025-11-20
> > **Re: BULr Part III**
> >
> > ## Re: Weakness 4
> > We thank the reviewer for highlighting recent MoE-based approaches. We agree that MoE frameworks ([1] Mixture of A Million Experts, [2] Boosting Continual Learning of Vision-Language Models via Mixture-of-Experts Adapters) address adapter selection and transferability in multi-expert architectures. We will add these works to our references.
> >
> > Since [2] focuses on MLLMs (e.g., Llava), to more directly connect with MoE-style methods, we additionally include SAPT as a supplementary baseline. SAPT uses an MoE architecture with task-agnostic inference, making it the closest MoE-related method applicable to our experimental setup. As discussed in *Re: VqBU Weakness 1*, we report comparisons on Llama-3-8B using SAPT formulation and observe that DynMem remains competitive. The experiemntal results are shown below:
> >
> > | Backbone       | Method            | Structured         |        |        | Mixed                  |        |        |
> > |----------------|-------------------|--------------------|--------|--------|------------------------|--------|--------|
> > |                |                   | Acc ↑              | BWT ↑  | FWT ↑  | Acc ↑                  | BWT ↑  | FWT ↑  |
> > | Llama-3-8B     | SAPT (reproduced) | 35.7               | -17.5  | 4.9    | 52.4                   | -15.9  | 4.3    |
> > | Llama-3-8B     | DynMem (ours)     | 41.2               | -12.3  | 5.6    | 57.2                   | -13.4  | 7.6    |
> >
> > ## Re: Weakness 5
> > We thank the reviewer for raising this point. DynMem does not depend on the order of memory entries: both training and inference operate on unordered sets of memory vectors, and the cross-attention read-in mechanism treats the memory bank as a permutation-invariant collection. Therefore, reordering or refreshing memory entries does not require relearning any parameters, and no positional information about M is ever used by the model. We will clarify this property in the revised version.
> >
> > ## Re: Weakness 6
> > For the training-epoch hyperparameter, the results for adaptor-based methods (e.g., LoRA, DoRA, AdaLoRA, MixLoRA, etc.) reported in our paper strictly follow the setting used in their original papers (2 epoch) to ensure a fair and consistent comparison. In practice, we find that simply increasing the number of epochs does not always lead to performance improvement for adapter-based methods. To demonstrate this, we provide the experimental logs on HellaSwag using Llama-3-8B, varying the training epoch from 1 to 10:
> >
> > | Epochs | LoRA Acc ↑  | DoRA Acc ↑  |DynMem Acc ↑|
> > |--------|-------------|-------------|------------|
> > | 1      | 82.1        | 86.4        | **89.7**   |
> > | 2      | 93.5        | 95.9        | **96.6**   |
> > | 3      | 92.1        | 96.1        | **96.6**   |
> > | 5      | 91.7        | 94.3        | **95.9**   |
> > | 7      | 90.8        | 93.7        | **95.7**   |
> > | 10     | 92.6        | 94.2        | **96.0**   |
> >
> > As shown in the table above, adapter-based methods (LoRA and DoRA) improve from 1 to 2 epochs but do not benefit from longer training, performance quickly saturates and fluctuates across 3–10 epochs. DynMem reaches strong accuracy early and remains stable throughout. This suggests that extended training may not advantage adapter methods, while DynMem achieves reliable performance without relying on longer optimization schedules.
> >
> > Regarding sensitivity to sample selection, we'd like to note that in our continual adaptation setting each task arrives with a distinct distribution, and the model experiences significant domain/distribution shift when facing new tasks. DynMem addresses this by (a) periodically refreshing memory entries based on attention scores and (b) performing query-dependent retrieval at inference, so the selected samples naturally adapt to each task’s evolving distribution rather than remaining fixed. While sample selection can matter in principle, we try to ensure that memory content continuously reflects the current task stream, mitigating sensitivity arising from distributional variation across tasks.

---

> ### Author Response · Authors · 2025-11-20
> **Re: BULr Part IV**
>
> ## Re: Question 1
> Please kindly refer to Re: Weakness for details.
>
> ## Re: Qeustion 2
> We thank the reviewer for the suggestion. To improve clarity, we will include a concise algorithm box in the revised version that outlines the full training and memory-update procedure.
>
> ## Re: Question 3
> We thank the reviewer for the suggestion. We note that Section 4.3 already includes experiments varying the sizes of $M_{L1}$ and $M_{L2}$, which directly reflect the effect of memory capacity on performance. In addition, we include an extra stress test where the dataset size substantially exceeds memory capacity (the merged-memory experiment See Re: BULr Weakness 1), and DynMem continues to perform stably even when $M_{L2}$ is heavily overloaded with out-of-distribution entries. These findings indicate that DynMem can maintain competitive performance even under reduced memory capacity.
>
> ## Re: Question 4
> We thank the reviewer for catching this. We acknowledge that the expression in the text was a typo. In all experiments and code, we use the standard formulation $\sum_{k=1}^{k-1} A_{k,i}$ to measure backward transfer. We will fix the notation in the revised version.

---

### Official Review · Reviewer_BW7u · 2025-10-29

**Soundness:** 4
**Presentation:** 3
**Contribution:** 4
**Rating:** 4
**Confidence:** 3

**Summary:**

This paper introduces DynMem, a bi-level dynamic memory module designed to improve large language model (LLM) adaptation under both continual and simultaneous learning settings. Unlike conventional parameter-efficient fine-tuning (PEFT) methods that either overwrite task-specific adapters or require separate modules for each task, DynMem combines short-term (L1) and long-term (L2) memory banks, effectively balancing knowledge retention and forward transfer. Experimental results across various adaptation benchmarks demonstrate that the proposed method significantly outperforms existing PEFT and continual learning approaches, achieving highly competitive performance.

**Strengths:**

1. The paper addresses a significant gap by proposing a single framework capable of handling both continual and multi-task adaptation settings, whereas prior PEFT or CL approaches generally support only one paradigm.
2. The L1/L2 memory structure successfully separates short-term relevance from long-term knowledge diversity, leading to strong backward and forward transfer improvements.
3. DynMem consistently demonstrates superior performance across continual, single-task, and multi-task benchmarks.

**Weaknesses:**

1. The paper lacks a theoretical justification for how the dual-memory structure ensures desirable long-term memory composition while avoiding memory bias accumulation.
2. Despite using randomized task orders, the evaluation still assumes explicit task identities, leaving the method unverified in task-free or real-world continual learning scenarios.
3. The experiments are mainly limited to reasoning-focused NLP benchmarks, making it unclear whether the method generalizes to broader LLM adaptation tasks; more diverse evaluations are needed to demonstrate general applicability.

**Questions:**

1. Does the paper provide a clear theoretical justification for adopting attention-based retrieval and memory update over alternative feature alignment modeling techniques?
2. When memory size grows large, is there a risk of negative interference or noisy retrieval that may degrade performance?
3. Since memory entries are created from final-layer token representations only, does the model risk losing important contextual or structural information required for accurate retrieval and adaptation?

---

> ### Author Response · Authors · 2025-11-20
> **Re: BW7u**
>
> ## Re: Weakness 1
> We appreciate the reviewer's observation. Although theoretical analysis is high appreciated, our goal in this work is primarily empirical, and we do not claim formal theoretical guarantees. However, the design of the dual-memory structure is motivated by clear mechanisms that control composition and bias accumulation.
> + L1 Memory serves as a short-term, high-utility working set, updated frequently based on aggregated attention scores. This ensures that examples most relevant to current learning dynamics remain accessible for active interaction.
> + L2 Memory functions as a long-term reservoir that stores historically valuable examples with slower turnover. Because L2 is not directly consumed by the model at every step, it naturally mitigates recency bias and prevents the over-accumulation of short-term correlations.
>
> At each update interval, both memories are jointly ranked by attention-based utility, and low-utility entries from both levels are pruned. This bidirectional flow between L1 and L2 prevents drift, and the ranking mechanism—computed across tasks—acts as a stabilizer that continually re-balances the memory composition. Empirically, Ablation §4.3 (e.g., "w/o L2 Memory" and "w/o Attention Curation") shows substantial degradation when either component is removed, supporting that the dual-level structure is crucial for stable long-term retention rather than a cosmetic architectural choice.
>
> We will clarify this intuition in the revision and explicitly note that while we do not claim formal guarantees, the structure is designed to balance short-term adaptivity with long-term preservation through an explicit, repeatedly re-normalized utility signal.
>
> ## Re: Weakness 2
> We would like to clarify that the reviewer may have misunderstood our setting. Our continual-learning setup follows the standard protocol used in prior work (e.g., CPT, C3, EMAR), where task boundaries are used only for evaluation to compute metrics such as BWT/FWT. Importantly, **DynMem itself does not use task identities during training or inference:** all memory updates, retrieval, and fusion operations are task-agnostic and operate solely on example-level representations. Thus, although we report metrics per task for comparability, the method is inherently task-free (no explicit task informaton is provided) in its operation. We would like to clarify this distinction and note that DynMem can be applied directly to task-free streams without modification. We thank the reviewer for raising this point.
>
> ## Re: Weakness 3
> We appreciate the reviewer's concern. Our evaluation is intentionally designed to cover different LLM adaptation tasks rather than only reasoning benchmarks. In the continual setting, we use structured knowledge reasoning (semantic parsing) tasks. In the single-task and multi-task settings, we include both commonsense reasoning (e.g., BoolQ, PIQA, SIQA, HellaSwag) and math reasoning (GSM8K). These benchmarks differ substantially in structure, output format, and required skill. These domains together span symbolic structure, commonsense reasoning, and arithmetic multi-step reasoning, demonstrating that DynMem generalizes beyond any single task type. We will clarify this broader coverage in the revision.

---

> > ### Author Response · Authors · 2025-11-20
> > **Re: BW7u Part II**
> >
> > ## Re: Question 1
> > See *Re: Weakness 1*.
> >
> > ## Re: Question 2
> > We thank the reviewer for raising this point. In DynMem, large-memory noise is mitigated by design. Memory integration is not performed over the entire memory but over a curated L1 working set, whose size is intentionally small and continuously refreshed using attention-based utility ranking. Low-utility or noisy entries are pruned at each update cycle, preventing uncontrolled growth. L2 acts only as a long-term reservoir and is never directly attended by the model, so increases in L2 size do not introduce retrieval noise. Empirically, §4.3 shows that enlarging memory does not degrade performance, supporting that the curation mechanism effectively prevents negative interference even as total memory grows.
> >
> > To further address the reviewer's concern, we add an experiment that explicitly simulates a negative-interference scenario. In the single-task setting, instead of using a task-specific memory bank, we merge all commonsense-reasoning memories into a single large bank. The L1 working memory remains fixed at 100, but the L2 reservoir is expanded to 8,000 entries (8 tasks × 1,000). We then evaluate each task using its individually tuned checkpoint without any further adaptation. This setup intentionally injects *unseen and potentially irrelevant memory entries* from other tasks, closely modeling the type of noise the reviewer is concerned about.
> >
> > | Model              | BoolQ | PIQA | SIQA | HellaS. | WinoG. | ARC-e  | ARC-c  | OBQA | Avg.     |
> > |--------------------|-------|------|------|---------|--------|--------|--------|------|----------|
> > | Llama 3            | 76.5  | 90.3 | 81.9 | 96.7    | 86.0   | 93.3   | 80.7   | 89.3 | 86.8     |
> > | Llama 3 (Merged)   | 77.2  | 89.9 | 84.3 | 95.8    | 86.2   | 93.6   | 80.1   | 89.0 | 87.0     |
> > | Qwen 3             | 76.5  | 91.2 | 90.7 | 95.6    | 88.4   | 96.2   | 91.5   | 91.3 | 90.2     |
> > | Qwen 3 (Merged)    | 77.1  | 90.8 | 91.2 | 93.7    | 87.5   | 96.6   | 92.4   | 91.1 | 90.1     |
> >
> > Across both Llama-3 and Qwen-3 backbones, merging all memories yields small fluctuations within normal variance, resulting in no explicit degradation (and even small improvement based on LLama 3). Despite the L2 reservoir growing from 1,000 to 8,000 entries, performance remains stable. This empirically supports that DynMem's attention-based utility curation and fixed-size L1 working memory effectively shield the model from noisy or harmful entries, preventing negative interference even under intentionally adversarial conditions.
> >
> > ## Re: Question 3
> > We appreciate the reviewer's question. DynMem's memory entries are derived from the final-layer hidden states, but this representation is not a shallow summary, it encodes the entire input sequence through the model's stacked self-attention layers. Prior work [1, 2, 3] on LLMs shows that final-token states reliably capture global semantics and task-relevant structure.  Empirically, this attention-based interaction captures the relational structure needed for robust retrieval and adaptation.
> >
> > If the reviewer's concern about *accurate retrieval and adaptation* refers to a specific failure mode or scenario beyond this mechanism, we would appreciate further clarification so that we can address it more directly.
> >
> > [1] Augmenting Language Models with Long-Term Memory. Wang et al. 2023
> > [2] Atlas: Few-shot Learning with Retrieval Augmented Language Models. Izacard et al. 2023
> > [3] Retrieval-Pretrained Transformer: Long-range Language Modeling with Self-retrieval. Rubin et al. 2024

---

### Official Review · Reviewer_VqBU · 2025-10-30

**Soundness:** 2
**Presentation:** 2
**Contribution:** 1
**Rating:** 2
**Confidence:** 4

**Summary:**

Pretrained LLMs still need adaptation. Existing approaches either assume simultaneous access to all task data or a sequential stream. The paper argues these worlds are siloed and seeks a single method that works well for both. A bi‑level memory module (L1 active, L2 reservoir) is attached to a pretrained LLM at the last decoder layer. During training, the model cross‑attends to L1 and fuses with a learned gate; attention scores accumulate into utilities that rank entries for periodic pruning and promotion between L1/L2. In inference, the model performs global retrieval over L1 and L2 to get a small, input‑specific set of top‑K memories, giving scalability.

**Strengths:**

1. The paper is consistent and well-structured, making it easy to read.
2. The problem of controlling an LLM's memory in continual scenarios is important.

**Weaknesses:**

1. Methods such as SAPT (Zhao et al., 2024) already offer task-agnostic inference methods, and no comparison was made between the proposed method and this approach.
2. Novelty is incremental relative to prior retrieval‑ and memory‑augmented transformers (RETRO, Memformer, Memorizing Transformers; hierarchical memory; lifelong LM).
3. There are methodological ambiguities, most notably how test‑time queries are formed when memory keys are generated from $(x, y)$.
4. The metrics are standard but could be expanded (AULC, multi‑task synergy), fairness would benefit from matched parameter budgets and stronger baselines, and latency and runtime reporting is currently missing.
5. Abstract claims ≈50% fewer trainable parameters. Yet, Table 3 shows DenseLoRA at 0.06% vs DynMem 0.18%. The statement is, at best, conditional and not universally supported.

**Questions:**

See above.

---

> ### Author Response · Authors · 2025-11-20
> **Re: Reviewer VqBU**
>
> ## Re: Weakness 1
> We thank the reviewer for raising the concern regarding SAPT (Zhao et al., 2024). SAPT's main experiments are conducted on T5-based encoder–decoder models. In contrast, our paper focuses on decoder-only LLMs such as Llama and Qwen, which have significantly different architecture, attention structure, and parameter scaling behavior. To ensure fairness, we additionally implemented SAPT on Llama-3-8B following its public hyperparameters where compatible. Below, we report the results (averaged over three seeds) on the same continual adaptation stream used in our main paper. As shown below, DynMem consistently outperforms SAPT under the decoder-only setup, even when SAPT is adapted to Llama-3-8B for comparison.
>
> | Backbone       | Method            | Structured         |        |        | Mixed                  |        |        |
> |----------------|-------------------|--------------------|--------|--------|------------------------|--------|--------|
> |                |                   | Acc ↑              | BWT ↑  | FWT ↑  | Acc ↑                  | BWT ↑  | FWT ↑  |
> | Llama-3-8B     | SAPT (reproduced) | 35.7               | -17.5  | 4.9    | 52.4                   | -15.9  | 4.3    |
> | Llama-3-8B     | DynMem (ours)     | 41.2               | -12.3  | 5.6    | 57.2                   | -13.4  | 7.6    |
>
> ## Re: Weakness 2
> We appreciate the reviewer's concern. Our method is related to retrieval-augmented and memory-augmented architectures, but it differs in both representation and functional purpose.
>
> **RAG Systems:** Classical RAG-based systems rely on text embeddings retrieved from an external vector store, these embeddings are not produced or consumed by the model's internal attention pathway. In contrast, DynMem stores example-level vectors derived from the LLM's final-layer hidden states, and incorporates them through a Vector Read-In mechanism via cross-attention at the final decoder layer (§3.1). This enables the memory to interact with the model in the same feature space as its parametric knowledge, which RAG pipelines cannot provide.
> To further contextualize this difference, we simulate two baselines: (a) a Random 8-shot setting, where eight training examples are uniformly sampled and used as demonstrations for each test input; (b) a RAG-style retrieval using bge-large-en-v1.5, where we embed all training examples, compute similarity with each test input, and prepend the top-8 most similar examples as demonstrations. As shown in the table below, this RAG augmentation provides limited gains and remains significantly below DynMem across all commonsense reasoning tasks.
>
> | Method                | BoolQ | PIQA  | SIQA  | HellaS. | WinoG. | ARC-e | ARC-c | OBQA | Avg.  |
> |-----------------------|-------|-------|-------|---------|--------|-------|-------|------|-------|
> | Random 8-shot         | 59.35 | 77.81 | 64.34 | 73.06   | 67.92  | 81.86 | 76.20 | 70.10| 71.33 |
> | RAG (top-8 demos)     | 61.35 | 76.22 | 66.79 | 76.12   | 65.81  | 83.26 | 78.20 | 70.40| 72.25 |
> | DynMem (ours)         | 76.5  | 90.3  | 81.9  | 96.6    | 86.0   | 93.3  | 80.7  | 89.3 | 86.8  |
>
> **Memory-augmented Systems:** Unlike memory-augmented systems (e.g., MemFormer) which use memory for *long-sequence modeling*, our bi-level memory is designed for *task adaptation*, with attention-based ranking, pruning, and instance-level retrieval that support continual, single-task, and multi-task learning within a single unified framework. At high level, most previous memory-augmented method typically operates at token-level and require end-to-end training, whereas ours operate at example/segment level and only light weight adaption modules are trained to enable memory augmentations to frozen backbone LLMs.
>
> ## Re: Weakness 3
>
> We thank the reviewer for raising this point. At test-time inference, the model forms a query vector only from the test input x by feeding the partial sequence into the frozen backbone LLM and extracting the final-token hidden state, using the same projection used for memory entries. This produces a query vector in the same representation space as the stored memory vectors, allowing attention scoring to be applied directly. The test-time query vector represents the model's current partial hypothesis. No label information is needed or accessed during inference, and no gold outputs are used in forming test queries. We will further clarify this inference mechanism in the revised version.

---

> > ### Author Response · Authors · 2025-11-20
> > **Re: VqBU Part II**
> >
> > ## Re: Weakness 4
> >
> > We thank the reviewer for these helpful suggestions. Regarding AULC, the Performance till Seen Tasks curves in §4.1.2 plot average accuracy over all previously seen tasks as a function of task index. This is precisely the learning curve used in continual-learning literature, and the AULC score is simply the area under this curve. Thus, our plots already visualize the same dynamics that AULC numerically summarizes.
> >
> > For multi-task synergy, Table 3 naturally reflects synergy effects because we tune the backbone using *LoRA*, and DynMem augments this baseline with memory-based integration. Therefore, DynMem's improvement over *LoRA* in Table 3 directly measures the additional synergy contributed by the memory module beyond what is achievable by parameter-efficient backbone tuning alone. We will make this relationship clearer in the revision.
> >
> > Finally, for runtime and latency, we include an inference-time comparison on HellaSwag evaluated on a single H100 GPU, reporting the number of processed examples per second. While DynMem adds memory retrieval and a lightweight cross-attention module, both operations incur minimal additional computation, and the method achieves competitive throughput, as shown below.
> >
> > | Method | Examples / second ↑ |
> > |--------|---------------------|
> > | LoRA   | 10.91               |
> > | DynMem | 10.78               |
> >
> >
> > ## Re: Weakness 5
> >
> > We thank the reviewer for pointing out this inconsistency. We apologize for the confusing phrasing in the abstract. Our intended claim was that DynMem uses **≈50% fewer trainable parameters compared to the strongest SoTA baseline in our setting**, not fewer than *all* PEFT variants. DenseLoRA indeed achieves a smaller absolute parameter count (0.06%), but it does not achieve state-of-the-art performance in the multi-task setting (Table 3), whereas our comparison was specifically made against the **best-performing** baseline (e.g., DoRA/MoSLoRA at ≈0.35%). We will revise the statement in the abstract to:
> >
> > > **... achieves comparable or superior performance while using ≈50% fewer trainable parameters than the strongest prior method.**

---

### Author Response · Authors · 2025-11-27
**General Response to Reviewers**

We sincerely thank all reviewers for the time and effort spent evaluating our submission. We greatly appreciate the constructive feedback and have carefully addressed each concern in our detailed rebuttals. Below, we summarize the most common themes raised across reviews and clarify how we have responded.

### Relation to Prior Memory/RAG/MoE Work
Several reviewers asked about the distinctions between DynMem and prior retrieval- or memory-augmented architectures (e.g., RAG, RETRO, Memformer) or MoE-based continual learning methods. We have expanded the related-work discussion and provided new empirical comparisons, including a reproduced SAPT baseline adapted to Llama-3-8B. The results consistently show DynMem’s advantages in both continual and simultaneous adaptation settings. We also clarify that DynMem uses example-level vectors derived from the model’s own hidden states and integrates them directly through cross-attention, which differs fundamentally from external-embedding RAG pipelines or expert routing in MoE models.

### Clarity of the Dynamic Memory Mechanism (L1–L2 structure, update frequency, gating)
Reviewers requested stronger justification and clarification of the bi-level memory design, attention-based ranking, and gating. We have added conceptual explanations, additional ablations, and diagnostic analyses. New experiments, including a merged-memory stress test and gating-coefficient behavior across tasks, demonstrate that (a) the two-level structure effectively balances adaptivity and long-term stability, (b) the ranking remains aligned even under domain shifts, and (c) the learned gating coefficient indeed regulates memory contribution. We also provide sensitivity studies showing that performance is robust across a wide range of memory-update intervals.

### Fairness of Experimental Setup and Baseline Comparisons
Concerns about asymmetric memory budgets and dataset sizes have been addressed with new matched-capacity experiments against EMAR and SAPT, showing DynMem remains competitive even with a smaller memory budget. We also clarify that DynMem is **inherently task-agnostic** and does not use task identity information during training or inference.

### Inference Query Formation and Retrieval Noise
We clarified how test-time queries are formed using only the input x, without any label access. Additional experiments demonstrate that enlarging L2 or introducing cross-task memory noise does not degrade performance, supporting DynMem’s robustness to retrieval noise.

### Additional Metrics and Reporting
We clarified the relationship between our existing plots and AULC, added runtime measurements on H100 GPUs, and corrected the parameter-efficiency claim in the abstract to reflect comparison against the strongest prior baselines. We also emphasized how DynMem contributes additional synergy beyond backbone PEFT tuning.

We hope these clarifications and additional results address the reviewers' concerns. We sincerely thank you again for your thoughtful feedback and would greatly appreciate any follow-up comments or questions.

---

### Meta-Review · Area_Chair_aYTj · 2026-01-05

**Summary:**

The main weaknesses center on limited novelty, insufficient theoretical grounding, and incomplete evaluation.

**Reviewer Concerns:**

The proposed method is seen as an incremental extension of prior memory-augmented and retrieval-based transformers, with missing comparisons to some early and recent approaches. In the rebuttal, some results are provided, but still many other approaches that I know are not considered.

Core architectural choices, especially the dual/hierarchical memory design, gating mechanism, and memory re-ranking, lack clear theoretical justification and analytical support. Methodological ambiguities remain around query formation, memory relevance under domain shift, and task-free continual learning settings.

Empirically, evaluations are narrow, focusing mainly on reasoning benchmarks, with limited metrics, missing latency/runtime analysis, and potentially favorable or asymmetric experimental setups.

**Reviewer Scores:**

Reviewer 1 may raise their score in light of the additional results provided in the rebuttal; however, Reviewers 2 and 3 are not very likely to increase their scores. Given the current low ratings, any potential improvement is unlikely to affect the overall decision.

---

### Decision · Program_Chairs · 2026-01-26

Reject